# Optimal Sequence and Second-Line Systemic Treatment of Patients with *RAS* Wild-Type Metastatic Colorectal Cancer: A Meta-Analysis

**DOI:** 10.3390/jcm10215166

**Published:** 2021-11-04

**Authors:** Chih-Chien Wu, Chao-Wen Hsu, Meng-Che Hsieh, Jui-Ho Wang, Min-Chi Chang, Ching-Shiang Yang, Yi-Chia Su

**Affiliations:** 1Division of Colorectal Surgery, Department of Surgery, Kaohsiung Veterans General Hospital, Kaohsiung 813, Taiwan; pauleoswu@gmail.com (C.-C.W.); cwhsu@vghks.gov.tw (C.-W.H.); wang@vghks.gov.tw (J.-H.W.); mcchang100@vghks.gov.tw (M.-C.C.); 2Department of Surgery, College of Medicine, National Yang Ming Chiao Tung University, Taipei 112, Taiwan; 3Department of Hematology and Oncology, E-Da Cancer Hospital, Kaohsiung 824, Taiwan; philips115@gmail.com; 4College of Medicine, I-Shou University, Kaohsiung 824, Taiwan; 5Department of Health Business Administration, Meiho University, Pingtung 912, Taiwan; 6Department of Pharmacy, Kaohsiung Veterans General Hospital, Kaohsiung 813, Taiwan; csyang429@vghks.gov.tw; 7Institute of Clinical Pharmacy and Pharmaceutical Sciences, College of Medicine, National Cheng-Kung University, Tainan 701, Taiwan; 8Department of Nursing, Shu-Zen Junior College of Medicine and Management, Kaohsiung 821, Taiwan

**Keywords:** sequential therapy, patient management, *RAS*-WT mCRC, targeted therapy, metastatic cancer treatment

## Abstract

Although several sequential therapy options are available for treating patients with *RAS* wild-type (WT) metastatic colorectal cancer (mCRC), the optimal sequence of these therapies is not well established. A systematic review and meta-analysis of 13 randomized controlled trials and 4 observational studies were performed, resulting from a search of the Cochrane Library, PubMed, and Embase databases. Overall survival (OS) did not differ significantly in patients with *RAS*-WT failure who were administered a second-line regimen of changed chemotherapy (CT) plus anti-epidermal growth factor receptor (EGFR) versus only changed CT, changed CT plus bevacizumab versus changed CT plus anti-EGFR, or changed CT versus maintaining CT plus anti-EGFR after first-line therapy with CT, plus bevacizumab. However, OS was significantly different with a second-line regimen that included changed CT plus bevacizumab, versus only changing CT. Analysis of first-line therapy with CT plus anti-EGFR for treatment of *RAS*-WT mCRC indicated that second-line therapy of changed CT plus an anti-EGFR agent resulted in better outcomes than changing CT without targeted agents. The pooled data study demonstrated that the optimal choice of second-line treatment for improved OS was an altered CT regimen with retention of bevacizumab after first-line bevacizumab failure. The best sequence for first-to-second-line therapy of patients with *RAS*-WT mCRC was cetuximab-based therapy, followed by a bevacizumab-based regimen.

## 1. Introduction

Colorectal cancer (CRC) is a common malignancy worldwide and is one of the most common causes of cancer-related mortality [1]. Approximately 20% to 25% of CRC patients worldwide present with metastasis at the time of initial diagnosis [2]. Improvements in early detection and advances in comprehensive treatment have led to a reduction in mortality from metastatic colorectal cancer (mCRC) in recent years. In particular, the combination of chemotherapy (CT) with targeted monoclonal antibodies (mAbs) has received attention. Available options include bevacizumab, which binds to vascular endothelial growth factor (VEGF), and cetuximab and panitumumab, which act against the epidermal growth factor receptor (EGFR). Both types of mAbs have been approved for use as first-, second-, and even third-line treatment to improve survival in patients with mCRC [3,4,5,6,7]. Various chemotherapy regimens, such as FOLFOX (5-fluorouracil/leucovorin/oxaliplatin), FOLFIRI (5-fluorouracil/leucovorin/irinotecan), or FOLFOXIRI (5-fluorouracil/leucovorin/oxaliplatin/irinotecan), can be selected and combined with targeted biologics in patients with mCRC. However, the optimal sequence of systemic therapy for patients with different profiles is still not established, especially regarding second-line therapy.

A previous network meta-analysis demonstrated that first-line therapy combining anti-EGFR mAbs with chemotherapy was a more effective option for increasing overall survival (OS) in patients with rat sarcoma (*RAS*) gene wild-type (WT) mCRC than anti-VEGF mAbs combined with chemotherapy. However, there was no significant difference in progression-free survival (PFS) between treatments with anti-EGFR and anti-VEGF mAbs [8]. Bennouna et al. reported that treatment with bevacizumab was beneficial for patients with disease progression after first-line therapy with bevacizumab plus chemotherapy. Their study showed that continuing bevacizumab with chemotherapy (a regimen that had been a first-line treatment) as second-line therapy prolonged OS and PFS significantly, compared with chemotherapy alone [9]. However, another study revealed that first-line treatment with anti-EGFR therapy and crossover to later-line anti-VEGF therapy resulted in improved OS [10]. Therefore, the effects of the subsequent-line therapy choice after failed first-line therapy remain unclear.

Targeted agents and cytotoxic chemotherapy have been approved for use in different combinations, evident in various sequential therapy options (National Comprehensive Cancer Network, version 4, 2020). In addition, different sequences of chemotherapy and biological therapies can be provided to patients. However, the optimal sequence, or optimal choice of a second-line therapy, is not well established. In this systematic review and meta-analysis, we reviewed current evidence to identify the optimal choice of second-line systemic treatments in patients with *RAS*-WT mCRC.

## 2. Materials and Methods

### 2.1. Study Design

We conducted a systematic review and meta-analysis of randomized clinical trials (RCTs) and retrospective studies that compared the efficacy of bevacizumab, cetuximab, and panitumumab, combined with chemotherapy, as a sequence of therapies for patients with mCRC. This systematic review is reported in accordance with the Preferred Reporting Items for Systematic Reviews and Meta-Analyses (PRISMA) guidelines [11] and the Cochrane Collaboration format [12]. 

### 2.2. Search Strategy

The Cochrane Library, PubMed, and Embase databases were scrutinized for eligible articles from inception until August 2020. We applied automatic e-mail alarms via each database platform to recheck the databases for any relevant studies that were published up until the submission of this article. We designed and performed the search using keywords and medical subject heading terms without language restrictions, after consultation with a health science librarian in our institute. The search keywords were based on the following strategies: sequence or third or second, “metastatic colorectal cancer” or “advanced colorectal cancer”, and bevacizumab or cetuximab or panitumumab. Further details of the search strategy are shown in Appendix A. Studies were screened using titles, abstracts, and contents to determine whether they met the inclusion/exclusion criteria. Potentially relevant studies were retrieved as full texts and then assessed. Manual searches of the reference lists of each relevant report were carried out to determine if any relevant studies had been missed in the search strategy.

### 2.3. Study Selection

Eligible studies were required to meet the following inclusion criteria: (1) they included patients with mCRC; and (2) they evaluated sequence therapy where the first-line chemotherapy regimen was a combination that included bevacizumab, cetuximab, or panitumumab. Exclusion criteria were as follows: (1) studies without retrievable endpoints; (2) studies that were conducted on non-mCRC patients; and (3) studies that compared chemotherapy combined with bevacizumab, cetuximab or panitumumab, but not as first-line therapy, or monotherapy with bevacizumab, cetuximab or panitumumab alone. When several studies discussed the same trial, only the study with the most recent data was included.

### 2.4. Data Extraction

Two reviewers (C.-C.W. and Y.-C.S.) assessed the studies independently for eligibility, then extracted data from the eligible studies using a standardized data extraction form. Disagreements were resolved by discussion with a third author (C.-W.H.). The following parameters were extracted: study characteristics (first author, year of publication, country of study, study design, and study period); patient characteristics (Eastern Cooperative Oncology Group status); the number of patients in each treatment arm; characteristics of the treatment regimen (chemotherapy backbone, target therapy, first-, second-, or third-line); and the efficacy of each trial’s treatment regimen. We analyzed the OS, PFS, and objective response rate (ORR) for all patients. For reports of the same trial at different follow-up periods, data from the latest report were used in the analysis. Unpublished data from the included studies were obtained from the study authors.

### 2.5. Quality Assessment 

Quality assessment of the included RCTs was performed using the Cochrane Risk of Bias Tool 2.0 [13]. For each trial, a judgment of bias was provided on each of the following domains: allocation, performance, follow-up, measurement, reported bias, and overall. The retrospective studies included in the meta-analysis were assessed for methodological quality using the Risk Of Bias In Non-randomized Studies of Interventions (ROBINS-I) tool [14]. This tool assessed the risk of bias in estimates of the comparative effectiveness (harm or benefit) of interventions from studies that did not randomly allocate units (individuals or clusters of individuals) to comparison groups. The categories for risk of bias judgment are “Low risk,” “Moderate risk,” “Serious risk,” and “Critical risk” of bias. The overall risk from the retrospective studies that were judged as “Low risk” or “Moderate risk” on the ROBINS-I tool was included in further quantitative meta-analysis. These domains were judged by two reviewers (C.-C.W. and Y.-C.S.) as low, some concerns, or high, and any conflict was resolved through discussion with a third author (C.-W.H.).

### 2.6. Statistical Analysis

We conducted the meta-analysis using the DerSimonian and Laird random-effects model. The dichotomous outcome was used to calculate the overall odds ratio (OR) with a 95% confidence interval (CI) of the ORR. The survival outcome was considered the hazard ratio (HR) of the PFS and OS. If a multivariate analysis was reported, an adjusted HR was used. The inverse variance method was used to calculate the overall HR and the 95% CI [15]. Quantitative meta-analyses of the pooled effect estimates were calculated and presented using forest plots. Heterogeneity of the pairwise comparisons was measured using Cochran’s Q statistical test and I^2^ values. Statistical analyses were performed using Review Manager (RevMan) version 5.3.5 (Cochrane Informatics and Knowledge Management Department) for Windows [16]. Results were considered statistically significant when the *p*-value (two-sided) was <0.05.

## 3. Results

### 3.1. Characteristic Information of Search Results

From the search of the electronic databases, 1537 studies were identified. After reviewing the titles and abstracts, 1380 publications were determined to be either duplicates or irrelevant and were excluded. Of the remaining 157 articles retrieved for full-text evaluation, we excluded: 25 due to incorrect populations or non-relevant endpoints; 77 with inappropriate interventions; 36 that were reviews, commentaries, editorials, or protocol descriptions; and 2 that were papers describing the same trial. 

Finally, 13 RCTs and 4 retrospective studies were included in the meta-analysis (Figure 1). All of them reported outcomes with PFS, OS, and ORR in patients with Kirsten *RAS* (*KRAS*)-WT mCRC [3,4,5,6,7,10,17,18,19,20,21,22,23,24,25,26,27,28]. There were eight phase 3 studies of patients previously treated with VEGF-targeted agents that investigated the efficacy of various antiangiogenic agents or EGFR antibodies combined with the same or a different doublet chemotherapy backbone in a second-line setting [3,5,6,7,21,26,27,28]. Two studies reported the comparison of a second-line therapy with the same targeted agents using a different type of chemotherapy backbone, or different types of chemotherapy alone, in patients with mCRC who were treated with first-line therapy using an EGFR antibody [17,20]. One study compared the results of a second-line therapy with the same targeted agent and doublet chemotherapy, or a different targeted agent and doublet chemotherapy, in patients with mCRC who had received first-line therapy with an EGFR antibody and triplet chemotherapy (FOLFOXIRI) [18]. Five studies compared the results of a crossover in second-line therapy in patients with mCRC who had been treated with an EGFR antibody, or bevacizumab with chemotherapy, as first-line therapy [4,10,19,22,23,25]. The characteristics and measured effects of the 17 studies are summarized in Appendix A.

### 3.2. Risk of Bias Assessment

The quality of the eligible RCTs and retrospective studies was assessed using the Cochrane Risk of Bias 2.0 tool and the ROBINS-I tool, respectively. The overall risk of bias in the 13 RCTs was assessed using the Cochrane Risk of Bias 2.0 tool [3,5,6,7,10,17,18,21,24,25,26,27,28] (Appendix A). The overall risk of bias in the four retrospective studies were all judged as “Moderate risk” on the ROBINS-I tool and were included in the further quantitative meta-analysis [4,19,20,22] (Appendix A). 

### 3.3. Effect of the Different Sequence Regimens on PFS, OS, and ORR 

In patients with *KRAS*-WT mCRC, who had received first-line therapy with CT plus bevacizumab, the pooled HR of PFS and OR for the ORR indicated that second-line therapy with changed CT plus an anti-EGFR agent resulted in better outcomes, compared with patients receiving only a different CT (PFS, HR = 0.71, 95% CI: 0.56–0.92; ORR, OR = 2.96, 95% CI: 1.28–6.82). The OS did not differ significantly between the second-line therapy with changed CT plus anti-EGFR agent versus only changing the CT (OS, HR = 0.84, 95% CI: 0.62–1.14) (Figure 2A). The results for the PFS and OS showed a significant difference (PFS, HR = 0.61, 95% CI: 0.49–0.76; OS, HR = 0.69, 95% CI: 0.53–0.90), with second-line therapy using a different CT plus bevacizumab versus only a change in CT. The ORR did not differ significantly between the second-line therapy of changed CT plus bevacizumab versus a change in CT alone (ORR, OR = 1.56, 95% CI: 0.48–5.01) (Figure 2B). The analysis was performed by pooling the effects of second-line therapy using changed CT plus bevacizumab versus changed CT plus anti-EGFR agent on the PFS, OS, and ORR with first-line therapy, using CT plus bevacizumab in patients with *KRAS*-WT. The pooled results for the PFS, OS, and ORR indicated a non-significant benefit (PFS, HR = 0.92, 95% CI: 0.7–1.22; OS, HR = 0.91, 95% CI: 0.69–1.20; ORR, OR = 0.34, 95% CI: 0.11–1.07) (Figure 2C). In patients with *KRAS*-WT who had first-line therapy with CT plus bevacizumab, the results indicated that second-line therapy with changed CT versus CT plus an anti-EGFR agent did not show a significant difference (PFS, HR = 1.03, 95% CI: 0.69–1.54; OS, HR=0.84, 95% CI: 0.55–1.28; ORR, OR = 1.73, 95% CI: 0.80–3.73).

In patients with *KRAS*-WT who had first-line therapy with CT plus an anti-EGFR agent, the pooled HR for the PFS and OR for ORR indicated that second-line therapy with changed CT plus an anti-EGFR agent resulted in better outcomes, compared with only changing the CT without targeted agents (PFS, HR = 0.74, 95% CI: 0.59–0.94; OS, HR = 0.76, 95% CI: 0.59–0.99; ORR, OR = 2.21, 95% CI: 0.95–5.17) (Figure 2D). 

The analysis was then performed by pooling the effects of CT plus an anti-EGFR agent crossover with CT plus an anti-VEGF agent, versus CT plus an anti-VEGF agent crossover to CT plus an anti-EGFR agent on the OS of patients with *KRAS*-WT. The pooled results for OS showed a significant benefit (OS, HR = 0.70, 95% CI: 0.58–0.83) whereas the PFS and ORR did not show significant differences (PFS, HR = 0.78, 95% CI: 0.47–1.28; ORR, OR = 0.91, 95% CI: 0.60–1.38) (Figure 2E).

(A) Second-line therapy of change chemotherapy (CT) and targeted agent (TG) versus change in first-line therapy with bevacizumab plus chemotherapy.

### 3.4. Efficacy of Primary Tumor Sidedness and Crossover Regimens on the OS

The analysis was performed by pooling the efficacy of CT plus an anti-EGFR agent crossover to CT plus an anti-VEGF agent, versus CT plus an anti-VEGF agent crossover to CT plus anti-EGFR agent on the OS of patients with *KRAS*-WT who had different tumor localization. The pooled results for the OS showed a significant benefit when the tumor was located on the left side (OS, HR = 0.70, 95% CI: 0.57–0.85). There was no significant difference when the tumor was located on the right side (OS, HR = 0.90, 95% CI: 0.49–1.66) (Appendix A).

## 4. Discussion

The comparison of “switching chemotherapy backbone with bevacizumab” and “switching chemotherapy, but without targeted therapy” as second-line therapy while the disease progressed after first-line therapy with bevacizumab plus chemotherapy showed that PFS and OS outcomes were better with “switching chemotherapy backbone with bevacizumab” (Figure 2B) [6]. In the comparison between “switching both chemotherapy backbone and targeted therapy” as second-line therapy and “switching chemotherapy, but without any targeted therapy,” PFS was significantly better; however, there was no significant difference in the OS (Figure 2A) [7,26,27]. In the comparison between “switching chemotherapy backbone but keeping bevacizumab” as second-line therapy and “switching chemotherapy backbone and targeted therapy to an anti-EGFR mAb,” a non-significant survival outcome was shown (Figure 2C) [3,21,28]. A similar result for the survival outcome was observed in a comparison between “only switching chemotherapy” to “keeping chemotherapy but switching targeted therapy to an anti-EGFR” [5]. Contrastingly, statistically significant differences in OS were observed in patients who had been treated with first-line therapy comprising an anti-EGFR mAb or an anti-VEGF mAb plus chemotherapy, followed by a crossover in the second-line therapy, regardless of the type of chemotherapy backbone (Figure 2E) [4,10,22,25]. The above analyses also found worse survival outcomes in patients treated with cetuximab following bevacizumab. The biological mechanisms that have the potential to reduce the efficacy of anti-EGFR antibodies when administered after antiangiogenic therapy are not clearly understood. In vitro studies using *RAS*-WT mCRC tumor cells have emphasized that an anti-VEGF mAb may activate the *RAS* pathway, promoting resistance to anti-EGFR antibodies [29]. Moreover, overexpression of VEGF-A induced by bevacizumab was found to be involved in acquired resistance to anti-EGFR antibodies [30,31,32]. The aforementioned analysis for a subsequent choice of therapeutic regimen may help establish a second-line therapy after the failure of first-line treatment with bevacizumab, thus achieving better survival. 

In patients with mCRC who had been treated with cetuximab plus chemotherapy as first-line therapy, “switching chemotherapy backbone but keeping cetuximab” as second-line choice showed better survival and ORR than “switching chemotherapy, but without any targeted therapy” (Figure 2D) [17,20]. However, cetuximab plus chemotherapy as first-line therapy followed by a second-line regimen of bevacizumab plus chemotherapy, regardless of the chemotherapy backbone, showed better OS than bevacizumab-based therapy as a first-line regimen and cetuximab-based therapy as a second-line regimen (Figure 2E) [4,10,22,25]. This suggests that keeping cetuximab while switching the chemotherapy regimen is the optimal second-line regimen after failure of the first-line cetuximab-based therapy.

Subgroup analysis based on primary tumor sidedness found that cetuximab plus chemotherapy as first-line therapy, followed by a second-line regimen with bevacizumab plus chemotherapy, led to better survival outcomes in left-sided *KRAS*-WT mCRC cases. For right-sided mCRC, there was no significant difference in either crossover sequence. These results are consistent with those of a previous network meta-analysis on first-line therapy for mCRC [8].

First-line therapy is well known for its key role in the successful treatment of patients with mCRC, which is attributed to its long treatment duration. It is most effective in regard to objective response and PFS. Therefore, choosing an effective first-line therapy is important, especially for patients with potentially resectable metastases [33]. However, subsequent therapy is indicated for a small number of patients who may have reduced performance and reduced tolerance to adverse effects. Nonetheless, if curative resection is not achieved despite first-line therapy, the sequence and choice of the subsequent line of therapy may be critical. Once the therapeutic sequence has been established, we still need to determine the optimal therapeutic approach. This study aimed to compare switching chemotherapy, not only within treatment lines but also switching the chemotherapy regimen and targeted agent.

Our study had a few limitations. First, it included an analysis of summary statistics rather than individual patient data. This may have resulted in the presence of covariates that affected treatment outcomes, especially in patients who received diverse later-line therapy, different sequences of biological agents, and interventions for metastases. We were unable to estimate the impact of these confounding factors on patient outcomes. Second, the disease severity of patients with mCRC varied and included those with oligometastatic disease, initial resectability of distant metastasis, and medical comorbidities. The evolution of metastasectomies and non-surgical interventions, whose outcomes are dependent on surgical techniques across studies, may have resulted in a variety of outcomes. Third, the pooled data were without a fixed first-line chemotherapy regimen. However, our analysis focused on whether to switch the first-line chemotherapy backbone to a second-line regimen.

## 5. Conclusions

Our meta-analysis identified the therapeutic efficacy of different targeted therapy sequences, especially first- and second-line therapies, in patients with *RAS-*WT mCRC. Furthermore, it explored the impact of switching the chemotherapy backbone and the primary tumor sidedness in different sequences of first- and second-line therapies. Our pooled data study demonstrated that the optimal choice of second-line therapy to improve OS was a change in chemotherapy regimen with the retention of bevacizumab after first-line bevacizumab failure. Aside from this, the best sequence for first-to-second-line therapy for patients with *RAS*-WT mCRC was cetuximab-based therapy, followed by a bevacizumab-based regimen.

## Figures and Tables

**Figure 1 jcm-10-05166-f001:**
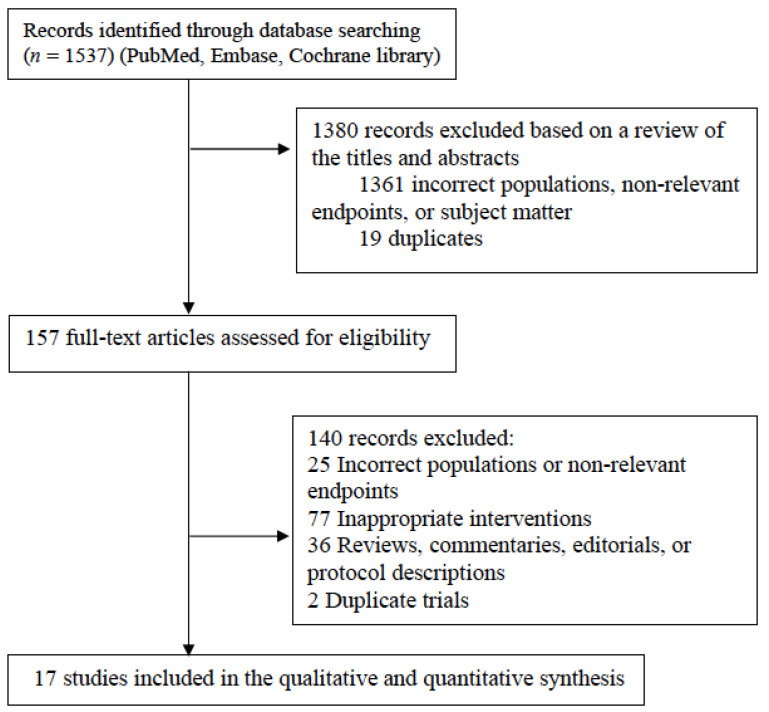
Preferred Reporting Items for Systematic Reviews and Meta-Analyses (PRISMA) guidelines flowchart summarizing study identification and selection.

**Figure 2 jcm-10-05166-f002:**
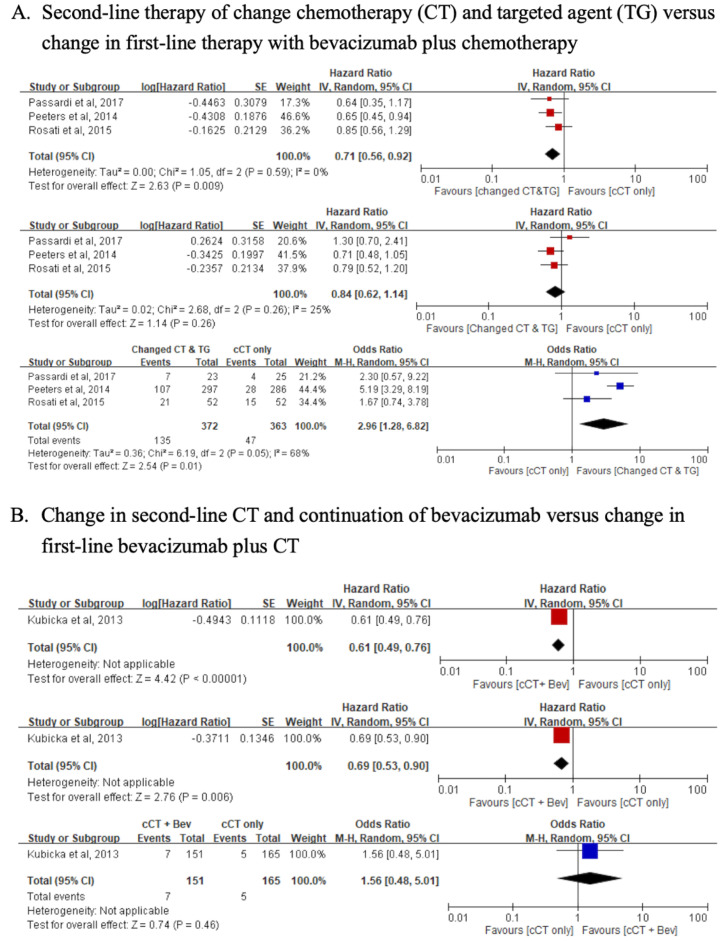
Effect of the different sequence regimens on the progression-free survival, overall survival, and objective response rate. (**A**) Second-line therapy of change chemotherapy (CT) and targeted agent (TG) versus change in first-line therapy with bevacizumab plus chemotherapy; (**B**) Change in second line CT and continuation of bevacizumab versus change in first-line bevacizumab plus CT; (**C**) Change in second-line CT and continuation of bevacizumab versus change CT and TG to anti-EGFR Ab in first-line bevacizumab plus CT; (**D**) Change CT and continuation of cetuximab versus change CT only in first-line anit-EGFR Ab plus CT; (**E**) First- and second-line targeted therapy with anti-VEGF Ab and anti-EGFR Ab crossover; Abbreviations: CT, chemotherapy; cCT, changed chemotherapy; TG, targeted agent; Bev, bevacizmab; SD, standard deviation; VEGF, vascular endothelial growth factor; EGFR, epidermal growth factor receptor; IV, interval variable; CI, confidence interval; Ab, antibody; M-H, Mantel–Haenszel.

## Data Availability

All relevant data to this study are available by contacting the corresponding author.

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
