# Peer review of "Optimal Sequence and Second-Line Systemic Treatment of Patients with RAS Wild-Type Metastatic Colorectal Cancer: A Meta-Analysis"

_jcm, 2021, doi:10.3390/jcm10215166_

Round 1

Reviewer 1 Report

This is a well-written systematic review analyzing the sequence of therapies for RAS wild-type metastatic colorectal cancer. However, I have some major concerns about the systematic review methodology especially about how studies were included for quantitative analysis. These issues need to be clarified prior to publication.

The search was conducted over a year ago to August 2020, but authors do state that databases were rechecked for relevant studies up to the publication date. However, this is not sound PRISMA methodology as the search should be redone for these missing dates and studies should be rescreened. This was not clearly stated in the methodology or shown in the PRISMA flow chart.

Search strategy included the terms "sequence or second or third". This may potentially miss articles that did not include these terms in their titles or abstracts but still included data on therapeutic sequence. Was a health science librarian consulted to perform the literature search? This was not mentioned in the methodology.

Why was the Cochrane Risk of Bias tool used for RCTs but the Newcastle-Ottawa used for the retrospective studies? I would suggest using Cochrane risk of bias tools for both as there is a specific Robins-I tool for non-randomized studies. See link below. This would lead to more consistency since both assessments will be using Cochrane-based tools. Additionally, the Newcastle-Ottawa scale has demonstrated quite poor inter-rater reliability and is not the best risk of bias tool to use.

https://pubmed.ncbi.nlm.nih.gov/22855629/

https://methods.cochrane.org/bias/risk-bias-non-randomized-studies-interventions

The study inclusion is confusing. In the abstract, you mentioned that 5 RCTS and 3 observational studies were included yet the results section and Figure 1 mention 17 studies were included for both qualitative and quantitative synthesis. This needs to be clarified. Perhaps 17 studies were included for qualitative and only 8 were included for quantitative analysis?

On line 186 it says that “Data from the retrospective studies that scored from 7 to 8 on the Newcastle–Ottawa Scale criteria were included in the quantitative analysis”

However, this analysis strategy was not mentioned in the methods section. Better methodology would include an analysis with all relevant data. A further subgroup/sensitivity analysis that includes only the higher quality studies can then be performed. This way, the reader can be sure that studies are not being specifically chosen to achieve a significant finding.

Author Response

Response to Reviewer 1 Comments

Point 1: The search was conducted over a year ago to August 2020, but authors do state that databases were rechecked for relevant studies up to the publication date. However, this is not sound PRISMA methodology as the search should be redone for these missing dates and studies should be rescreened. This was not clearly stated in the methodology or shown in the PRISMA flow chart.

Response 1: Thanks for your kind suggestions regarding the unclear description of the search method for updated studies. We have clarified it in the Materials and Methods Section [Page 2, lines 92-93].

Point 2: Search strategy included the terms "sequence or second or third". This may potentially miss articles that did not include these terms in their titles or abstracts but still included data on therapeutic sequence. Was a health science librarian consulted to perform the literature search? This was not mentioned in the methodology.

Response 2: Thanks for your kind opinion. When designing the key term and search strategy, we have not only discussed with a health science librarian, but also referred to other similar reviews. It should be the most appropriate term and strategy in our opinion. We have clarified it in the Materials and Methods Section [Page 2, lines 93-94].

Point 3: Why was the Cochrane Risk of Bias tool used for RCTs but the Newcastle-Ottawa used for the retrospective studies? I would suggest using Cochrane risk of bias tools for both as there is a specific Robins-I tool for non-randomized studies. See link below. This would lead to more consistency since both assessments will be using Cochrane-based tools. Additionally, the Newcastle-Ottawa scale has demonstrated quite poor inter-rater reliability and is not the best risk of bias tool to use.

Response 3: Thanks for your kind suggestions regarding the suitable tool for evaluating the risk of bias for non-randomized control studies. We have re-assessed these studies by your recommended tool and revised it in the Materials and Methods Section [Page 3, lines 133-140], Results Section [Page 6, lines 190-192], and Supplementary Table 3.

 Point 4: The study inclusion is confusing. In the abstract, you mentioned that 5 RCTS and 3 observational studies were included yet the results section and Figure 1 mention 17 studies were included for both qualitative and quantitative synthesis. This needs to be clarified. Perhaps 17 studies were included for qualitative and only 8 were included for quantitative analysis?

Response 4: Thanks for your delicate suggestions regarding the miswrote numbers of enrolled studies in the abstract. We have revised it in the abstract [Page 1, lines 24-25].

Point 5: On line 186 it says that “Data from the retrospective studies that scored from 7 to 8 on the Newcastle–Ottawa Scale criteria were included in the quantitative analysis”

However, this analysis strategy was not mentioned in the methods section. Better methodology would include an analysis with all relevant data. A further subgroup/sensitivity analysis that includes only the higher quality studies can then be performed. This way, the reader can be sure that studies are not being specifically chosen to achieve a significant finding.

Response 5: Thanks for your kind suggestions regarding the unclear description of the risk of bias assessment and the selection strategy for further quantitative meta-analysis. We have changed the assessment tool for non-randomized studies and revised it in the Materials and Methods Section [Page 3, lines 138-140], and Results Section [Page 6, lines 190-192].

Reviewer 2 Report

Authors Chih-Chien Wu et al., provided an insightful and useful meta-analysis about optimal sequence and second-line systemic treatment of patients with RAS 22 wild-type (WT) metastatic colorectal cancer (mCRC). They collected the large amount of data from the Cochrane Library, PubMed, and Embase etc., total 25 databases, that was sufficient and reliable. Based on these data, they identified the best therapy for the patients with RAS-WT mCRC as cetuximab-based therapy, followed by a bevacizumab-based 37 regimen among five clinical trails and three observational studies. Moreover, they also explored the impact of switching the chemotherapy backbone and the primary tumor sidedness in different sequences of first- and second-line therapies, this would be extremely useful strategy for clinical application. The present manuscript is of high scientific quality content and, overall, clear to the reader. The experimental design is clear and robust, and correlated with a strong statistical analysis. I suggested this MS only need minor revise, such as the graphic clarity (Fig. 2). 

Author Response

Response to Reviewer 2 Comments

Point 1: Authors Chih-Chien Wu et al., provided an insightful and useful meta-analysis about optimal sequence and second-line systemic treatment of patients with RAS 22 wild-type (WT) metastatic colorectal cancer (mCRC). They collected the large amount of data from the Cochrane Library, PubMed, and Embase etc., total 25 databases, that was sufficient and reliable. Based on these data, they identified the best therapy for the patients with RAS-WT mCRC as cetuximab-based therapy, followed by a bevacizumab-based 37 regimen among five clinical trails and three observational studies. Moreover, they also explored the impact of switching the chemotherapy backbone and the primary tumor sidedness in different sequences of first- and second-line therapies, this would be extremely useful strategy for clinical application. The present manuscript is of high scientific quality content and, overall, clear to the reader. The experimental design is clear and robust, and correlated with a strong statistical analysis. I suggested this MS only need minor revise, such as the graphic clarity (Fig. 2).

Response 1: Thanks for your kind suggestions regarding the graphic clarity of Figure 2. We have improved the resolution of Figure 2.

Reviewer 3 Report

The manuscript is very interesting although it has some limitations which however do not significantly alter the results and conclusions of the manuscript itself. Good statistical analysis and English. Discussion and references can be improved also on the basis of conclusions already expressed in previous articles on the topic (for example "Surgery for colorectal cancer in elderly: a comparative analysis of risk factor in elective and urgency surgery" Boselli, C et al; or "Analysis of long-term results after liver surgery for metastases from colorectal and non-colorectal tumors: A retrospective cohort study" Parisi, A).

Author Response

Response to Reviewer 3 Comments

Point 1: The manuscript is very interesting although it has some limitations which however do not significantly alter the results and conclusions of the manuscript itself. Good statistical analysis and English. Discussion and references can be improved also on the basis of conclusions already expressed in previous articles on the topic (for example "Surgery for colorectal cancer in elderly: a comparative analysis of risk factor in elective and urgency surgery" Boselli, C et al; or "Analysis of long-term results after liver surgery for metastases from colorectal and non-colorectal tumors: A retrospective cohort study" Parisi, A).

Response 1: Thanks for your kind recommendation. Based on the association of our topic, we appreciated and cited one of your recommended studies for improving the content of the Discussion Section [Page 2, lines 49-51, 53-58]